# Trainable subnetworks reveal insights into structure knowledge organization in protein language models

Ria Vinod[1], Ava P. Amini[2], Lorin Crawford[2]*, Kevin K. Yang[2]*

**1** Center for Computational and Molecular Biology, Brown University, Providence, Rhode Island, United States of America, **2** Microsoft Research, Cambridge, Massachusetts, United States of America

* lcrawford@microsoft.com (LC); yang.kevin@microsoft.com (KKY)

## Abstract

Protein language models (PLMs) pretrained via a masked language modeling objective have proven effective across a range of structure-related tasks, including high-resolution structure prediction. However, it remains unclear to what extent these models factorize protein structural categories among their learned parameters. In this work, we introduce trainable subnetworks, which mask out the PLM weights responsible for language modeling performance on a structural category of proteins. We systematically trained 39 PLM subnetworks targeting both sequence- and residue-level features at varying degrees of resolution using annotations defined by the CATH taxonomy and secondary structure elements. Using these PLM subnetworks, we assessed how structural factorization in PLMs influences downstream structure prediction. Our results show that PLMs are highly sensitive to sequence-level features and can predominantly disentangle extremely coarse or fine-grained information. Furthermore, we observe that structure prediction is highly responsive to factorized PLM representations and that small changes in language modeling performance can significantly impair PLM-based structure prediction capabilities. Our work presents a framework for studying feature entanglement within pretrained PLMs and can be leveraged to improve the alignment of learned PLM representations with known biological concepts.

### Author summary

Proteins govern cellular processes and their functions arise from the three-dimensional structures encoded by their amino acid sequences. Predicting protein structure from sequence has thus become a central capability of modern biological sequence models. Protein language models, trained on sequence alone with a general language modeling objective, are remarkably accurate at structure prediction and are widely used in protein design and engineering workflows.

**Data availability statement:** All code is available under an open-source MIT license at https://github.com/microsoft/plm_subnetworks. Training data, model configurations and checkpoints, and results are available via the link in the repository.

**Funding:** The author(s) received no specific funding for this work.

**Competing interests:** The authors have declared that no competing interests exist.

However, relatively little is known about how these models' weights encode relationships between different protein structural features. This direction is increasingly important as protein language models scale in data, compute, and model size. Here, we demonstrate that it is possible to isolate subsets of model weights, i.e., subnetworks, that correspond to specific categories of defined structures. Our results show that the structure-prediction accuracy using protein language models is highly sensitive to these subnetworks, even when changes in language modeling performance are small. When applied across diverse structural categories, our method suggests that structural knowledge is distributed in a way that reflects the continuous spectrum of protein structural diversity. Our work provides insight into how biologically relevant information is organized within protein language model weights and offers a foundation for a more informed and interpretable way to train future models.

## Introduction

Understanding protein structure is essential for deciphering biological function, as a protein's structure governs its molecular stability, interactions, and activity. Recent protein language models (PLMs) trained purely on sequence data have been shown to learn representations that implicitly encode structural information [1–4]. These models have proven effective in a wide range of protein engineering tasks including structure prediction [5–7], function annotation [8,9], mutation effect estimation [10], and even the design of novel proteins [11–13].

Many PLMs are pretrained with a self-supervised masked language modeling (MLM) objective, where models are tasked with predicting the amino acid identities of randomly masked tokens in a sequence [1,2,14,15]. Since protein structure is fundamentally determined by amino acid sequence, PLMs can implicitly encode structural information in their weights. ESM-2, a family of protein language models trained on evolutionary-scale data, showed that performance on the language modeling task can predict the quality of a PLM-predicted structure [1,7]. PLM representations are now widely used as inputs to models that predict structural coordinates [16], and scaling analyses have shown that improvements in MLM performance improve single-sequence structure prediction accuracy [7,17].

The simplicity and efficacy of pretraining language models on protein sequences have led to growing interest in understanding their internal mechanisms, establishing interpretability as an important research direction. With respect to the structure features that PLMs learn, prior work has shown that protein contact information is stored in attention maps [18], that PLMs learn co-evolving motifs which emulate an understanding of fundamental biophysics [19], and that sparse latent features from PLMs capture known functional biophysical properties and motifs [20,21].

To date, however, it remains unclear (i) if and how structure information is factorized and stored in the learned PLM weights, and (ii) whether this factorization affects performance on downstream structure prediction tasks. One approach to uncover whether such factorization of concepts in pretrained language models exist is via

subnetwork discovery [22]. Subnetworks are sparse computational subgraphs of a pretrained model's weights that are responsible for performance on a specific task or class of inputs. In the natural language domain, subnetwork discovery has been widely used to uncover linguistic properties—such as semantics, syntax, and relational entities—learned during pretraining by defining concepts and localizing weights, neurons, or layers that encode them [23–27].

In this work, we focus on leveraging subnetworks to interrogate the factorization of structural categories of proteins in the pretrained PLM, ESM-2 (Fig 1). Our goal was to find PLM subnetworks—sparse subgraphs of the original model weights—that when isolated, suppress the model's ability to make correct predictions on one class of inputs while preserving MLM performance on all other classes of inputs. As part of our main analysis, we systematically trained 39 such PLM subnetworks to suppress either residue- or sequence-level structural information across different scales of the CATH hierarchy. Our results reveal that structural categories are indeed encoded in a factorized manner within PLM weights and that, although non-suppressed inputs achieve ESM-2-level perplexity, structure prediction is still statistically significantly perturbed. Together, our subnetworks approach and the suite of analyses we perform provide insight into how PLMs organize structural features among learned parameters.

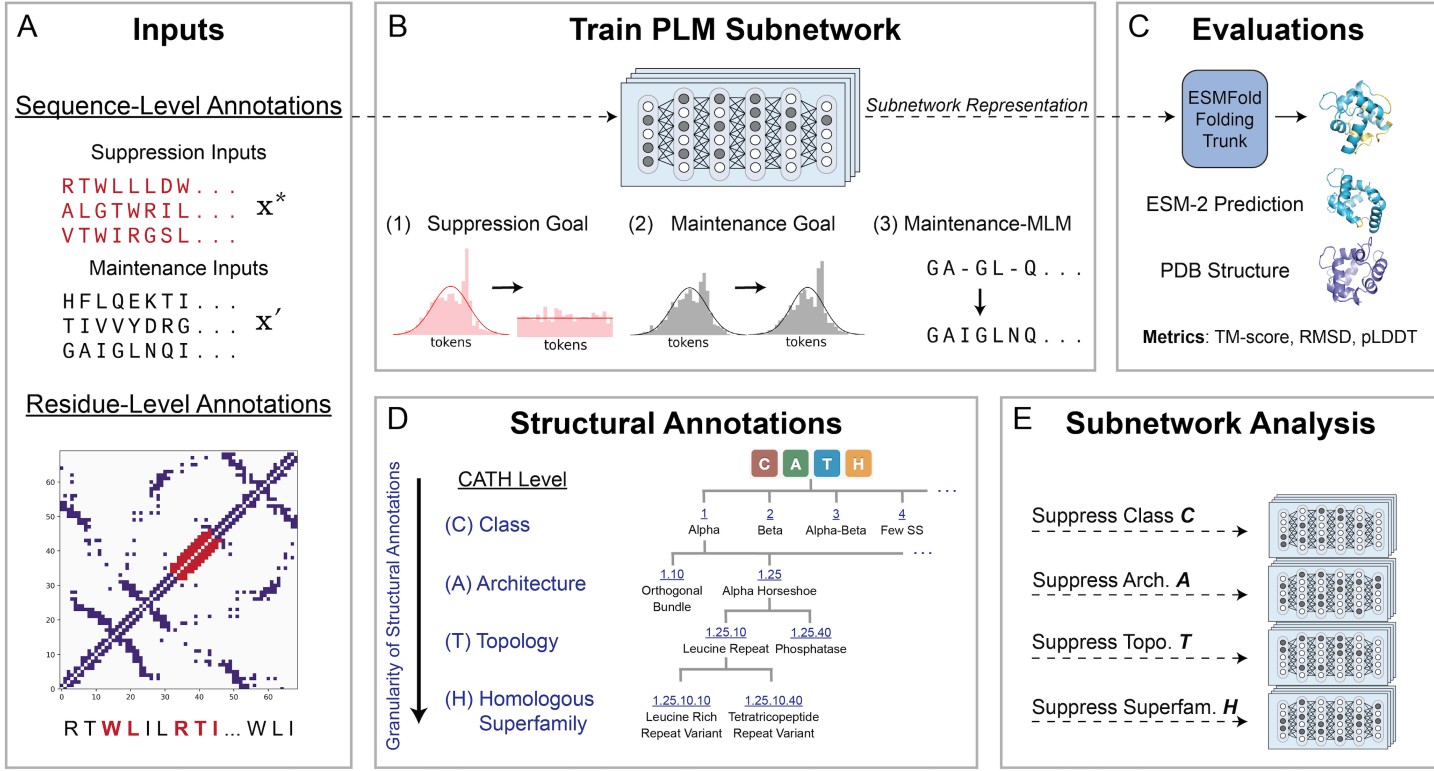

**Fig 1**. **Summary of subnetwork training procedure and main set of evaluations and analysis. (A)** Suppression inputs can be defined at either the protein sequence-level (based on CATH annotations) or the residue-level (based on 3-way secondary structure annotations). Suppressed inputs are red; maintained inputs are black. **(B)** PLM subnetworks are trained via a weighted multi-objective differentiable mask learning scheme, which defines suppression and maintenance goals for a binary mask learning procedure. **(C)** The subnetwork sequence representation is passed as input to the ESM-Fold folding trunk to predict a 3D structure. The resulting 3D structure can be compared to the ESM-2 prediction for the same sequence, both of which are independently aligned to ground-truth structures from the Protein Data Bank (PDB). **(D)** The CATH database offers a hierarchical classification of protein structural annotations across four levels, grouping proteins by their fold patterns and evolutionary relationships. Each successive level provides increasingly fine-grained structural annotations. **(E)** Independent subnetworks are trained to suppress different structural categories.

## Methods

### Suppression and maintenance inputs are defined by structural annotations

Protein secondary structure refers to the local spatial arrangement of amino acid residue backbone atoms. Each residue in a protein sequence adopts a secondary structure, which can be classified as an alpha helix, beta sheet, or loop. The resulting structural arrangements enable further categorization of proteins at the sequence level, grouping them by shared architectural and evolutionary features (Fig 1A). In this work, we define structural categories at the residue level (alpha helix, beta sheet, loop) using DSSP [28] and at the sequence-level (Class, Architecture, Topology, Homologous Super-family) using the CATH taxonomy (Fig 1D).

### Notation

We denote a pretrained PLM as $f(\mathbf{x}, \theta)$, where $\mathbf{x}$ represents the input sequence and $\theta$ are the model weights. The objective of our training procedure is to predict a binary mask, $\mathbf{m} \in \{0, 1\}^K$, where $K$ denotes the number of parameters in the PLM that we seek to learn the mask over. A subnetwork can be obtained by taking the Hadamard product between the binary mask and pretrained PLM weights, $f(\mathbf{x}, \mathbf{m} \odot \theta)$. Subnetworks are trained independently to suppress a structural category of sequence-level or residue-level inputs (i.e., all suppressed inputs belong to the same category defined by structural annotation). We define a suppression input sequence as $\mathbf{x}^*$ and all other input sequences (i.e., maintenance inputs) as $\mathbf{x}'$ (Fig 1A; suppressed inputs are red, maintained inputs are black). For residue-level suppression, let $\mathcal{A}$ be an annotation from DSSP that contains a set of $J$ positions corresponding to residues in {alpha helix, beta sheet, loop}. We define the $J$ suppression inputs for this DSSP annotation as $\mathbf{x}_{(J^*)}$. The maintenance inputs are all remaining complementary positions of residues with annotation $\mathcal{A}$, which are then defined as $\mathbf{x}_{(J')}$.

### Subnetwork training objective

To obtain a subnetwork $f(\mathbf{x}, \mathbf{m} \odot \theta)$, we learn a binary mask $\mathbf{m}$ using a weighted loss that consists of the following components (Fig 1B):

1. **Suppression goal.** If the mask is working appropriately, the subnetwork should struggle to reconstruct its suppressed inputs accurately. In other words, a well-calibrated predictive distribution over the suppressed input tokens should be approximately uniformly distributed with respect to the vocabulary $V$. As a result, we define the suppression loss as minimizing the Kullback–Leibler (KL) divergence between (i) the predictive distribution of the subnetwork over the tokens corresponding to the suppression inputs and (ii) a uniform reference distribution over tokens in the vocabulary (denoted as $\mathcal{U}_V$). For sequence-level suppression, this corresponds to

$$\mathcal{L}_{\text{supp}} = \text{KL}\left(f(\mathbf{x}^*, \mathbf{m} \odot \theta), \mathcal{U}_V\right). \tag{1}$$

   Similarly, for residue-level suppression,

$$\mathcal{L}_{\text{supp}} = \text{KL}\left(f(\mathbf{x}_{(J^*)}, \mathbf{m} \odot \theta), \mathcal{U}_V\right) \tag{2}$$

   where, again, the index $J^*$ is used to represent all the positions corresponding to residues with DSSP annotation $\mathcal{A}$.

2. **Maintenance-KL goal.** Even in the presence of a mask, the subnetwork should preserve the predictive behavior of the full PLM on maintenance inputs. Therefore, as a maintenance goal, we also aim to minimize the KL divergence between (i) the predictive distribution of the subnetwork over the tokens corresponding to the maintenance inputs and (ii) the predictive distribution for the pretrained PLM over the same elements. For sequence-level suppression,

this is expressed as the following

$$\mathcal{L}_{\text{maint}} = \text{KL}\left(f(\mathbf{x}', \mathbf{m} \odot \theta), f(\mathbf{x}', \theta)\right). \tag{3}$$

Similarly, for the residue-level suppression, we write

$$\mathcal{L}_{\text{maint}} = \text{KL}\left(f(\mathbf{x}_{(J')}, \mathbf{m} \odot \theta), f(\mathbf{x}_{(J')}, \theta)\right) \tag{4}$$

where, again, the index $J'$ denotes the set of positions not included in a given residue annotation $\mathcal{A}$ from DSSP.

3. **Maintenance masked language modeling (MLM) goal.** In practice, the maintenance-KL goal is insufficient because it only enforces similarity between the subnetwork and the original PLM output distributions, without ensuring that the subnetwork retains the ability to assign the correct probabilities to predicted tokens. Prior work demonstrated via a set of ablations that all 3 loss components are necessary to achieve the desired subnetwork behavior on suppression and maintenance inputs, and found that omitting either maintenance-KL or maintenance-MLM increased perplexity on the maintenance inputs [22]. We therefore introduce an additional maintenance-MLM loss, which ensures that the subnetwork can still allocate the appropriate probability mass to the right corresponding tokens, preserving its overall language modeling behavior. To include an MLM objective on the maintenance inputs, we randomly select 15% of sequence positions over which the MLM loss is computed. Of these $M$ positions, 80% are replaced with a mask token, 10% are randomly replaced, and 10% are unchanged. Namely,

$$\mathcal{L}_{\text{MLM}} = \sum_{i \in M} \log p_{\mathbf{m} \odot \theta}\left(x_i' \mid \mathbf{x}'_{(M')}\right) \tag{5}$$

where $M'$ are unmasked positions (i.e., true amino acid identities). To perform residue-level suppression, we again randomly choose a set of $M$ masked positions with the same mask-and-mutate scheme; however, this time, we compute the MLM loss with respect to the masked indices that overlap with the positions corresponding to a particular residue-level annotation $\mathcal{A}$, which we denote as $M \cap J$. This can be expressed as

$$\mathcal{L}_{\text{MLM}} = \sum_{i \in M \cap J'} \log p_{\mathbf{m} \odot \theta}\left(x_i \mid \mathbf{x}_{(M' \cap J')}\right) \tag{6}$$

where $M \cap J'$ denotes intersection of indices that are in $M$ and not in $J$, and $M' \cap J'$ represents the indices that are not in either set.

Overall, the final weighted training objective is then represented as the sum

$$\mathcal{L} = \lambda_1 \mathcal{L}_{\text{supp}} + \lambda_2 \mathcal{L}_{\text{maint}} + \lambda_3 \mathcal{L}_{\text{MLM}} \tag{7}$$

where $\lambda = \{\lambda_1, \lambda_2, \lambda_3\}$ are hyperparameters. A description of how we select training hyperparameters can be found in S1 Appendix.

## Differentiable weight masking for subnetworks

Following previous work [23,24,29], we adopt a differentiable weight masking scheme to learn $\mathbf{m}$. Here, each binary mask parameter $m_i \in \{0, 1\}$ is sampled from a Gumbel distribution. We learn a logit $l_i \in \mathbb{R}$ for every $i$-th parameter and obtain a

continuous mask score over the unit interval via the following Gumbel softmax transformation

$$s_i = \sigma\left\{-\frac{1}{\tau}\left[l_i + \log\left(\frac{U_1}{1 - U_1}\right)\right]\right\} \tag{8}$$

where $\sigma(\cdot)$ is the sigmoid function, $\tau$ is a temperature scaling hyperparameter, and $U_1 \sim \mathcal{U}(0, 1)$ is a random variable drawn from a standard uniform distribution. This Gumbel noise introduces stochasticity into the logit sampling process which results in exploring different binary mask configurations during training. We backpropagate through the collection of continuous mask scores **s** and threshold values to obtain binarized mask values **m** using the following

$$m_i = [\mathbb{1}(s_i > T) - s_i]_{\text{detach}} + s_i, \tag{9}$$

where $\mathbb{1}(\cdot)$ is an indicator function, $T$ is the mask score threshold, and $[\cdot]_{\text{detach}}$ prevents backpropagation through the discrete values in **m**. This thresholding operation enables differentiable training by allowing gradients to flow through the continuous mask scores, while still computing the loss with respect to binary predictions [26,30]. The sparsity is defined as the proportion of zeros in the learned binary mask—and therefore in the subnetwork—which is calculated as

$$\text{sparsity}(\mathbf{m}) = \frac{1}{K}\sum_{i=1}^{K}\mathbb{1}(m_i = 0), \tag{10}$$

where $\mathbf{m} = [m_1, \dots, m_K]^\top \in \{0, 1\}^K$ and, again, $K$ is the total number of mask parameters. In our results, we represent this number as a percentage and multiply the value above by 100%. A description of how we select weight masking hyperparameters can be found in the S1 Appendix.

### Model architecture and datasets

In our main set of analyses, we learned subnetworks in the ESM-2 650 million parameter model [7]. While other binary masking approaches often focus on the final layers to capture fine-grained concepts or properties, we chose to learn the mask over the full model, i.e., all layers, which prevents any reliance on weak early-layer signals and spurious late-layer correlations. For additional validation, we also apply our subnetworks approach to three state-of-the-art PLMs of varying size, architectures, and pretraining tasks: ProtBERT-UR100 [2], CARP-640M [14], and Dayhoff-170M-UR90 [31]. Details on the weight masking scheme, choice of which hidden layers to mask, hyperparameters, and mask initialization values for each of these models can be found in S1 Appendix).

For training and evaluation, we used the CATH S20 version 4.3.0 release [32]. This dataset is comprised of CATH domains, which are sequences clustered at a maximum of 20% pairwise sequence identity with at least 60% alignment overlap. This ensures low redundancy while maintaining structural and functional diversity across domains. Every domain has an annotation at successive levels of the CATH hierarchy: Class (C), Architecture (A), Topology (T), and Homologous Superfamily (H). We used DSSP [28] to obtain reduced three-way residue-level secondary structure categories, which is the procedure from [33]. During training, we filtered down to the set of CATH domains with length between 64 and 1024 residues (the maximum context window of ESM-2) and with available PDB structures. Altogether, this left 8886 CATH domains. For each subnetwork, we randomly split this data into 70% for training, 20% for validation, and 10% for a held out set. Subnetworks were learned according to the suppression and maintenance inputs present in the train split. At evaluation time, we performed MLM evaluations over all splits, which allowed for aggregating performance on suppression and maintenance inputs that are both seen during training via the train split, but also unseen inputs in the validation and test splits. We performed the structure prediction evaluations with the ESMFold folding trunk on the validation and test sets to limit computational cost. We provide details on CATH data and annotation frequencies in S1 Fig.

## Results

We used trained subnetworks to assess whether structural categories are factorized in PLM weights and, in turn, whether this factorization affects downstream structure prediction. To this end, we first evaluated 39 trained subnetworks to assess MLM performance on suppression and maintenance inputs. Second, we passed subnetwork representations as inputs to the ESMFold folding trunk to evaluate the effect of the suppression on structure prediction capabilities (see Fig 1C). In both sets of evaluations, we compared the effects of sequence-level suppression and residue-level suppression.

### Sparse subnetworks enable successful factorizations in ESM-2 weights

A structural category of proteins is considered to be factorized in PLM weights if we can identify a subnetwork that selectively reduces MLM performance on suppression inputs while maintaining performance on maintenance inputs, relative to the full PLM. We measured MLM performance using perplexity (i.e., the exponential of the negative log-likelihood), computed separately on the suppressed and maintained inputs for both the subnetworks and the pretrained ESM-2 model (Fig 2A).

By our training procedure, a subnetwork can achieve this differential performance by identifying a sparse subgraph of PLM weights where parameters that encode information about the suppressed input category are zeroed out. As the subnetwork training procedure does not explicitly promote any sparse regularization, the percentage of learned sparsity is intrinsic to any discovered implicit factorization. We quantified the mean learned sparsity (defined in Eq. (10) and reported as a percentage) across structural levels in Fig 2B. As suppressed categories become more fine-grained within the CATH hierarchy, the learned sparsity decreases, indicating that smaller structural categories can be factorized by zeroing out fewer parameters. This trend suggests a correlation between the learned percent sparsity of the subnetwork and the frequency of structural annotations (i.e., annotation granularity) in the data.

The trained subnetworks consistently yield higher perplexity on the suppression inputs, implying that structure-relevant representations can be selectively factorized in the PLM weight space (Fig 2C). Targeted suppression of residues belonging to a secondary structure (i.e., alpha helices or beta strands) results in an approximate two-fold increase in perplexity of these residues. Suppression of broader sequence-level annotations, such as CATH Class where the suppression set sizes are largest, yields predictive distributions resembling random noise, with perplexity approaching 20—the value expected from uniform guessing over 20 amino acids. A similar increase in perplexity is observed when suppressing Homologous Superfamilies, despite much smaller suppression sets (<100 sequences), suggesting that factorization is strongest at the coarsest and finest levels of annotation granularity. In contrast, structural categories are more weakly factorized at intermediate CATH levels like Architecture and Topology, as evidenced by lower perplexities on suppression inputs as compared to that of other CATH levels. One possible explanation for suppression being most effective at the Class and Homologous Superfamily levels is that these categories represent the most distinct structural and evolutionary boundaries in protein sequence space. Coarse Class categories capture global, fundamental secondary structure composition (i.e., mainly Alpha, mainly Beta, or "mixed" Alpha-Beta classes), while fine-grained superfamilies reflect localized evolutionary relationships, both of which have been shown to form well-separated regions in PLM embedding space [1,15,34]. In contrast, intermediate categories such as Architecture and Topology are structurally heterogeneous, combining folds that share only partial motifs but differ in geometry, making them inherently less separable and thus more difficult for subnetworks to isolate into distinct groups. When training is repeated across multiple ESM-2 seeds under the same suppression objective, the resulting subnetworks exhibit highly similar sparsity patterns and achieve nearly identical performance on both suppression and maintenance inputs, indicating that optimization converges to stable and functionally equivalent solutions (S2 Fig and S2 Table). The configurations of all trained subnetworks are reported in S3 Table, and the corresponding MLM performance is reported in S4 Table.

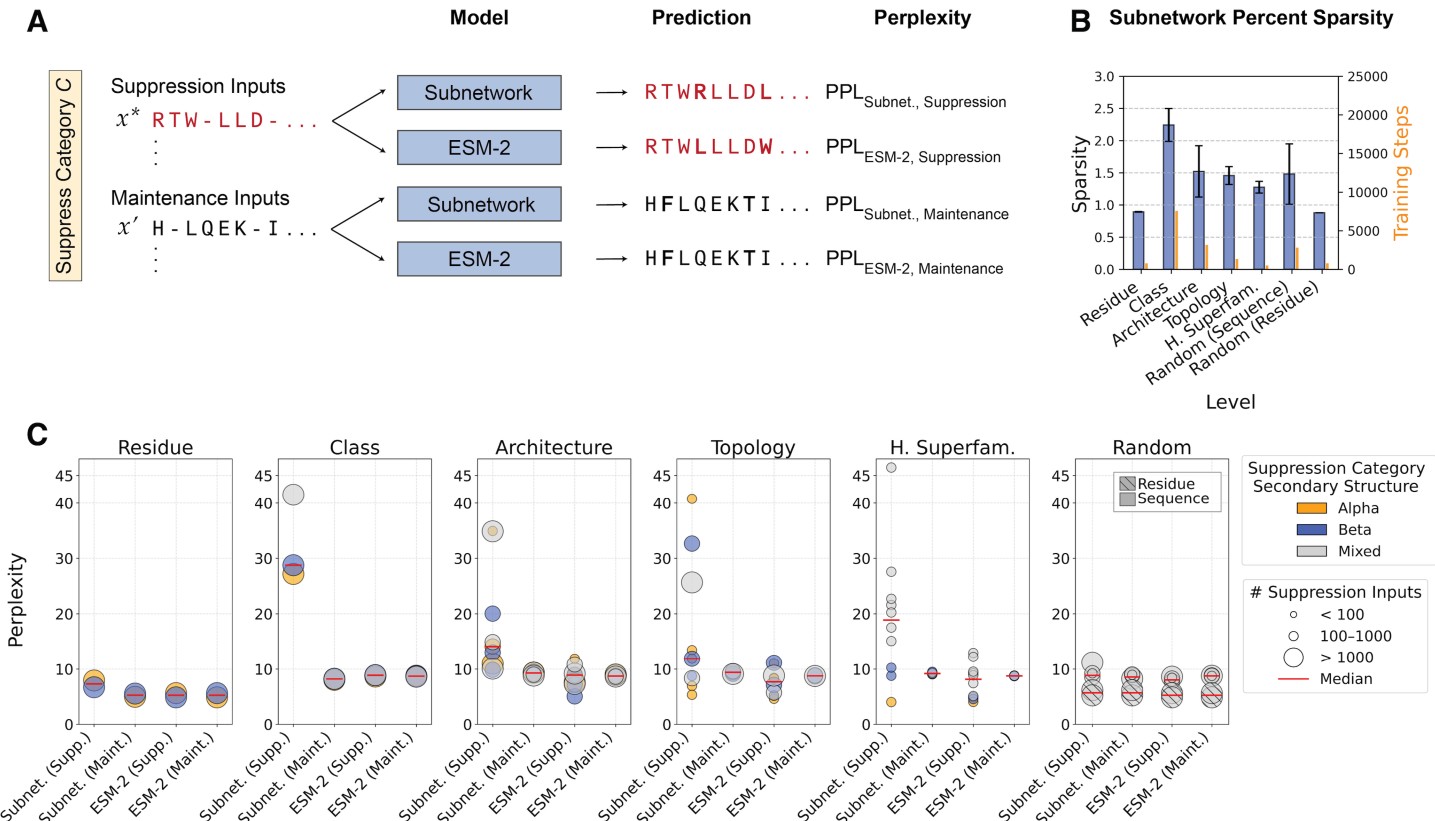

**Fig 2. Ablating less than 3% of ESM-2 parameters increases perplexity on suppressed inputs and reveals successful factorizations of structural categories in PLM weights. (A)** For a given subnetwork, suppression and maintenance inputs are defined by their structural annotation. At evaluation time, the same sequence is fed as separate inputs to the subnetwork and ESM-2. ESM-2 performance on the same suppression and maintenance inputs, labeled as "ESM-2 Supp." and "ESM-2 Maint." respectively, provides a baseline for the subnetwork. **(B)** Learned sparsity of each subnetwork across categories of structural annotations (reported mean with ± standard deviation depicted by the error bars). The right y-axis shows the average number of optimization steps in yellow. **(C)** Perplexity (y-axis) on test sequences across suppression and maintenance input categories (x-axis). Each point represents an independently trained subnetwork, where marker size indicates number of suppression inputs in the structural category and color indicates its predominant secondary structure type. Suppression and maintenance inputs are defined per subnetwork; ESM-2 points reflect baseline perplexity on those same inputs.

As a control, we trained two types of subnetworks. The first control was to suppress $N$ randomly selected sequences, where values of $N$ were chosen to mimic the size of a CATH sequence category. The second control was to suppress randomly selected residues. We evaluated the residue control subnetwork separately on alpha helices and beta sheets. Both the random sequence and random residue subnetworks fail to produce differential performance on the suppressed inputs, with the only exception being the sequence control with $N = 2000$, which resulted in a 2.6-point increase in perplexity on suppression inputs relative to ESM-2 (8.8). In contrast, subnetworks trained to suppress CATH-Class level categories of sequences are able to attain a perplexity of greater than 30, while achieving maintenance goals. We reasoned that any increase in perplexity in the sequence-controls arises because the mask learning process removes parameters that encode features broadly shared across sequences. Since randomly selected sequences do not share meaningful structural characteristics, the subnetwork cannot identify parameters specific to suppressed structural information and instead must zero out weights important for general MLM performance, which can lead to a uniform increase in perplexity across all inputs.

We also applied our subnetworks approach to three additional state-of-the-art PLMs of varying size, architectures, and pretraining tasks, each having been trained on UniRef data: ProtBERT-UR100 (420M parameters) [2], a transformer masked language model with a BERT architecture; CARP-640M (640M parameters) [14], a convolutional neural network masked language model where transformer layers are replaced by ByteNet dilated CNN blocks; and Dayhoff-170M-UR90 (170M parameters) [31], an efficient hybrid state-space-model transformer trained with an autoregressive objective. We present these analyses as additional results in S3 Fig. The subnetworks in these additional PLMs show similar trends to ESM-2, in that they are discovered by pruning less than 3% of model parameters, are most strongly factorized at the Class and Homologous Superfamily levels, and factorize categories of sequences to a larger extent when compared to residues.

Together, the percent sparsity and MLM evaluations show that subnetworks can increase perplexity in a targeted manner by identifying sparse factorizations of structural information in PLMs, aligned with the continuous nature of protein structural diversity.

### Subnetworks perturb structure prediction accuracy on both suppressed and maintained inputs

Having evaluated subnetworks on the language modeling task, we then assessed how suppression via a subnetwork affects structure prediction accuracy. ESMFold [7] introduced a folding trunk—a simplified version of AlphaFold2's Evoformer [35]—that converts language model representations into 3D structures. Using this frozen trunk allows the isolation of subnetwork-induced changes in sequence representations and enables investigation into how they affect structure prediction accuracy. Since the folding trunk serves as a fixed decoder, any degradation in template modeling (TM) score or predicted local distance difference test (pLDDT), or increased root mean square deviation (RMSD), directly reflects a loss of relevant structural information in the sequence representations. For each input sequence, we extracted sequence representations from the subnetwork and from ESM-2, and we passed them as separate inputs to the ESMFold folding trunk for structure prediction. Both predicted structures were independently aligned to the ground truth PDB structure to obtain a TM score, RMSD, and pLDDT for each model's prediction. This procedure was repeated on all suppression inputs $\mathbf{x}^*$ and all maintenance inputs $\mathbf{x}'$ in the validation datasets (Fig 3A).

First, we evaluated whether the subnetwork leads to a decrease in structure prediction accuracy relative to the ESM-2 performance on these inputs (Fig 3B). For each input, we computed the change in RMSD where $\Delta\text{RMSD} = \text{RMSD}_{\text{Subnet., PDB}} - \text{RMSD}_{\text{ESM-2, PDB}}$. These differences are consistently larger on suppression inputs and minor on maintenance inputs (Fig 3B). This suggests that the subnetwork is able to ablate structural information pertaining to the suppressed inputs in PLM weights. We then performed separate paired t-tests for the inputs in the suppression and maintenance sets, comparing $\text{RMSD}_{\text{Subnet., PDB}}$ and $\text{RMSD}_{\text{ESM-2, PDB}}$. Both tests result in significant $p$-values, suggesting that despite targeting only suppression inputs, the subnetwork also significantly affects structure prediction on maintenance inputs. That is, $\Delta\text{RMSD}_{\text{supp.}}$ and $\Delta\text{RMSD}_{\text{maint.}}$ are both statistically significant. We repeated this procedure using the TM-score and pLDDT and observed a similar trend (S4A and S4B Fig). We reason that the combination of pronounced effects on suppression inputs and minimal changes in structure prediction accuracy on maintenance inputs—together with the significant $p$-values—indicates that the frozen ESMFold trunk is sensitive to subtle representational shifts, rather than reflecting a lack of modularization in sequence representations. The subnetwork mask learning procedure was not optimized to preserve structure prediction accuracy; instead, it was designed to partition the PLM's weight space based on sequence-level characterization and MLM performance. Consequently, small variations in downstream structural metrics are expected, and this analysis mainly serves to reveal how sequence-level factorizations influence structure prediction capabilities.

Next, we assessed whether the magnitude of these differences is greater for suppression inputs than for maintenance inputs across CATH and secondary structure categories. That is, we seek to confirm that $|\Delta\text{RMSD}_{\text{supp.}}| - |\Delta\text{RMSD}_{\text{maint.}}| > 0$ (Fig 3C). These differences are consistently positive, suggesting that a subnetwork always results in a greater increase

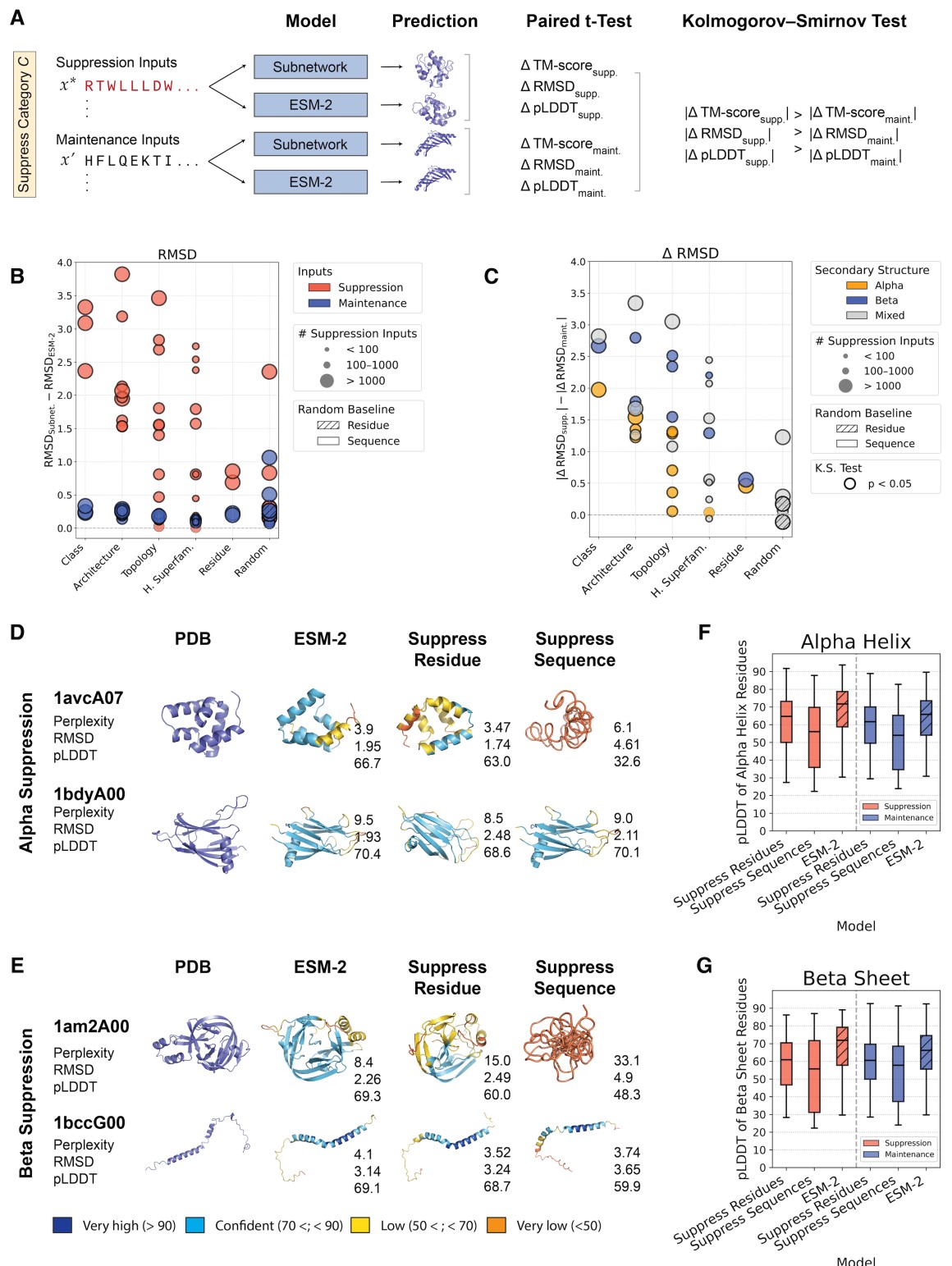

**Fig 3. Subnetworks consistently impair structure prediction capabilities on suppressed inputs. (A)** Sequence representations for a given input are obtained from the subnetwork and ESM-2 and fed as inputs to the ESMFold folding trunk to produce two predicted structures. Both structures are independently aligned to the ground truth PDB structure, and metrics are calculated for evaluations. **(B)** RMSD differences between the subnetwork

and ESM-2 baseline (y-axis) across structural levels (x-axis). Each point represents the RMSD increase for suppression inputs (red) or maintenance inputs (blue) relative to ESM-2. Marker size indicates the number of suppression inputs for each subnetwork. Bold outlines of markers indicate significant paired t-test p-values ($p < 0.05$). **(C)** Difference in absolute RMSD changes from the ESM-2 baseline (y-axis), stratified by structural level (x-axis). Each point corresponds to an individual subnetwork and shows the difference between the magnitudes of suppression and maintenance ΔRMSD values. Marker size reflects the number of suppressed inputs in the subnetwork, and color indicates secondary structure type. Bold outlines of markers indicate significance by Kolmogorov–Smirnov (KS) test ($p < 0.05$). **(D–E)** Visualizations and evaluation metrics for subnetwork predictions on select validation targets under (D) alpha suppression and (E) beta suppression. For each example, the ground truth PDB structure, ESM-2 prediction, and subnetwork predictions under residue-level and sequence-level suppression are shown. Each prediction is annotated with perplexity, mean RMSD, and mean pLDDT scores. **(F–G)** Distributions of predicted per-residue pLDDT scores across (F) alpha helix and (G) beta sheet residues across suppression and maintenance conditions, for subnetworks and baseline ESM-2 predictions. Each box shows the median and interquartile range.

in RMSD on the suppression inputs when compared to maintenance inputs. We also confirm that the differences in magnitude are consistently statistically significant by applying a two-sample Kolmogorov–Smirnov (KS) test between $|\Delta RMSD_{supp.}|$ and $|\Delta RMSD_{maint.}|$, rejecting the null hypothesis that the suppression and maintenance inputs exhibit the same distributions of difference in magnitudes of RMSD increase. We repeated the same analysis for TM-score and pLDDT and observed the same trend (S4C and S4D Fig). We also observe that the difference in magnitudes of change in each metric is, on average, larger for Alpha-Beta proteins, followed by Mainly Beta and then Mainly Alpha proteins. We reason that this could be due to stricter structural constraints imposed by the mixed secondary structure composition of Alpha-Beta folds, which couple alpha helices and beta sheets into interdependent architectures. Consistent with this interpretation, we used ProteinMPNN perplexities to evaluate sequence-level structural constraints and find a supporting trend: Alpha-Beta proteins exhibit the lowest perplexities, followed by Mainly Beta and then Mainly Alpha proteins, indicating that mixed Alpha-Beta folds have the most restricted sequence variability and strongest structure–sequence coupling (S5 Table). All structure prediction metrics and p-values are reported in S6 Table.

### Structure prediction is more sensitive to suppressed sequence features

Alpha helices and beta sheets are local secondary structure elements that, when predominant in a sequence, define the overall CATH Class of a protein domain. Sequences consisting of majority alpha helices form "Mainly Alpha" domains, while those rich in beta sheets form "Mainly Beta" domains at the CATH Class level. This connection allows us to directly compare the change in pLDDT between structures predicted by ESM-2 and those predicted by subnetworks trained to suppress (i) sequences labeled by CATH Class (e.g., Mainly Alpha or Mainly Beta), and (ii) residues belonging to specific secondary structures (e.g., alpha helices or beta sheets). We illustrated predicted structures for alpha suppression subnetworks in Fig 3D and beta suppression subnetworks in Fig 3E, where in each panel the top row is a suppressed input and the bottom row is a maintained input. While suppressing at the residue level based on secondary structure annotations only affects the mean pLDDT of the full predicted structure, suppressing sequences leads to no clear predicted fold or structure pattern. Since pLDDT is a per residue metric, we computed residue-specific pLDDT of predicted structures for both residue- and sequence-level suppression with alpha type (Fig 3F) and beta type (Fig 3G). Suppressing sequences still leads to a lower residue-specific pLDDT of the target residue type for both alpha and beta suppression subnetworks. This suggests that factorization of structural information in PLM weights is more sensitive to sequence-level suppression and, by extension, that PLMs more effectively model distributions at the sequence level.

### Discussion

We introduce subnetwork discovery as a post-hoc mechanistic approach to explore the modularity of learned representations in pretrained protein language models (PLMs). Because subnetworks are extracted without any fine-tuning, they expose relationships already present in the pretrained weights. The method is therefore a lightweight alternative to

PLOS Computational Biology

concept-level and other supervision-heavy interpretable pretraining schemes [36]. Our experiments show that PLMs can effectively disentangle both coarse and fine-grained CATH structural categories.

We chose ESM-2 for our main analyses because in addition to it being widely adopted and well-studied, it has an associated structure module that allows us to directly probe structure information in language modeling representations. We chose the 650M variant because it offers strong performance on both language modeling and structure prediction related tasks with reasonable computational cost. Learning a mask over model parameters effectively doubles GPU memory usage, and the 650M variant allowed us to train each subnetwork efficiently on a single H100 machine. Smaller ESM-2 variants yield less reliable structure prediction performance, while larger variants would be prohibitively expensive for the full set of experiments we presented. Since the performance of ESM-2 has been shown to scale with model size [7,17], we expect the subnetwork behavior observed here to exhibit similar scaling trends. In fact, our results offer a mechanistic explanation for this behavior: if structural information is organized within localized subsets of weights, then increasing model capacity expands the number of structural categories that can be represented, enhancing both representational and predictive performance. This interpretation is consistent with the view that PLMs encode statistics of co-evolving residues [19], as larger models trained on more diverse categories of sequences can store more subnetworks. We further find that subnetwork behavior is consistent across PLMs of different sizes, architectures, and training objectives, including ProtBERT-UR100 [2], CARP-640M [14], and Dayhoff-170M-UR90 [31]. This suggests that structural factorization is a general property of protein representation learning rather than an artifact of a specific model or training setup. The learned masks themselves provide further insight into where structural information is stored within a given PLM (S5–S8 Figs). In ESM-2, the subnetwork primarily zeros out parameters in mid-to-late layers (12–33), which aligns well with prior work showing that PLMs learn secondary structure features through superposition, with later layers becoming increasingly specialized for this purpose [20,21]. These layers likely correspond to higher-level abstractions of secondary and tertiary structure, analogous to how natural language models progressively learn linguistic hierarchy by encoding lexical and syntactic features in lower layers and semantic abstractions in higher layers [37,38].

Our training procedure operates directly in sequence space, and our results highlight the importance of broader sequence context in protein representation learning: we find that sequence-level categories are much more strongly factorized than residue-types. We found that while factorized PLM representations are similarly organized in the folding trunk weights of ESMFold, structure prediction accuracy is significantly perturbed when using representations from a subnetwork. The modest perturbations in structure prediction for maintenance inputs underscore that perplexity alone, typically the hallmark metric to evaluate PLMs, is not a sufficient measure of representational quality: even when a subnetwork reproduces ESM-2-level perplexity, small shifts in embedding distributions can propagate through the frozen ESMFold trunk and alter predicted structures.

Despite these successes, our work has some limitations. First, although comprehensive, our analysis is confined to the CATH dataset and to fundamental secondary structure categories. While assessing generalization to multi-domain proteins and complexes is an interesting future direction, the goal of this work is to characterize how structural categories are factorized in PLM weights at the level of their basic, clearly labeled secondary structure components. This approach relies on annotations which can be costly and time-consuming to curate, but are increasingly available with rapidly expanding with deep-learning based pipelines [39]. Second, our evaluation of structure prediction relies on the ESMFold folding trunk as the only available decoder capable of translating PLM representations into 3D coordinates. Since ESMFold is specifically designed for ESM-2 sequence representations, this restricts our ability to perform comparable analyses with other protein language models such as ProtBERT, CARP, and Dayhoff. Extending the folding trunk framework to models with different architectures or pretraining objectives would provide valuable insight into how these subnetwork representations directly influence structure prediction capabilities.

In sum, we envision that our approach will help accelerate PLM development and interpretability, as our framework can be readily extended to any labeled protein sequence dataset. It provides a lens to understand how factorizations in representation space affect downstream structural prediction tasks. As masked language model-based PLMs continue to scale, understanding how these models synthesize and organize biological information will become increasingly important.

## Supporting information

**S1 Appendix. Subnetwork training details.** Description of learning hyperparameters and compute.
(PDF)

**S1 Fig CATH annotation characteristics.** Three subnetworks were independently trained for each CATH Class suppression target (Mainly Alpha, Mainly Beta, Alpha-Beta) to assess the reproducibility of mask learning given random initialization of mask scores. Each point represents the validation perplexity of a subnetwork stratified by category of inputs. **(A) Annotation frequencies by CATH levels.** Each CATH domain is annotated with a label at the Class, Architecture, Topology, and Homologous Superfamily levels. Bar plots show the counts (y-axis) of the top 10 most frequent annotations (x-axis) at each level of the CATH hierarchy. **(B) Sequence length distributions stratified by CATH level.** Histograms show the counts (y-axis) of domain sequence lengths (x-axis) for each CATH Class: Mainly Alpha, Mainly Beta, Alpha Beta. **(C) Secondary structure composition of all CATH domains.** Average fraction of residues annotated as helix, strand, or coil across all CATH domains, based on DSSP annotations [28]. **(D) Secondary structure composition stratified by CATH class.** DSSP 8-state annotations are mapped to 3-state labels: H, G, I → H; E, B → E; T, S, - → L. Helix is H, beta strand is E, and loop is L.
(TIFF)

**S2 Fig ESM-2 CATH Class-level subnetwork language modeling performance across multiple seeds.** Three subnetworks were independently trained for each CATH Class suppression target (Mainly Alpha, Mainly Beta, Alpha-Beta) to assess the reproducibility of mask learning given random initialization of mask scores. Each point represents the validation perplexity of a subnetwork stratified by category of inputs.
(TIFF)

**S3 Fig Subnetwork language modeling performance on three additional PLMs of varying size and architectures.** To investigate whether subnetworks exist in pretrained PLMs other than ESM-2, we applied our approach to three additional models of varying size and architectures trained on UniRef data: ProtBERT-UR100, a transformer masked language model with a BERT architecture [2]; CARP-640M, a convolutional neural network masked language model (transformer layers are replaced by ByteNet dilated CNN blocks) [14]; and Dayhoff-170M-UR90, an efficient hybrid state-space-model transformer trained with and autoregressive objective [31]. **(A) ProtBERT-UR100 (420M).** Left: Learned percent sparsity by category of learned subnetworks in ProtBERT-UR100. Right: Masked language modeling performance on suppression and maintenance categories of sequences for each subnetwork. **(B) CARP-640M.** Left: Learned percent sparsity by category of learned subnetworks in CARP-640M. Right: Masked language modeling performance on suppression and maintenance categories of sequences for each subnetwork. **(C) Dayhoff-170M-UR90.** Left: Learned percent sparsity by category of learned subnetworks in Dayhoff-170M-UR90. Right: Autoregressive language modeling performance on suppression and maintenance categories of sequences for each subnetwork. Residue subnetworks were omitted from Dayhoff-170M-UR90 results because causal perplexity cannot be computed selectively over individual residues.
(TIFF)

**S4 Fig ESM-2 650M subnetwork-predicted TM-score and pLDDT with the ESM-2 folding trunk. (A–B)** Structural prediction differences (y-axis) between subnetworks and the ESM-2 baseline shown for **(A)** TM-score and **(B)** pLDDT across structural levels (x-axis). Each point represents change in metrics for suppression inputs (red) or maintenance

inputs (blue) for a subnetwork relative to ESM-2. Marker size indicates the number of suppression inputs for each subnetwork. Bold outlines of markers indicate statistically significant paired t-test $p$-values ($p < 0.05$). **(C–D)** Difference in absolute structure prediction metric changes from the ESM-2 baseline (y-axis), stratified by structural level, for **(C)** TM-score and **(D)** pLDDT. Each point corresponds to an individual subnetwork and shows the difference between suppression and maintenance ∆-values. Marker size reflects the number of suppressed inputs in the subnetwork, and color indicates secondary structure type. Bold outlines of markers indicate statistical significance by Kolmogorov–Smirnov (KS) test ($p < 0.05$).
(TIFF)

**S5 Fig Mask interpretation of ESM-2 650M.** Mean and standard deviation percent of parameters pruned by layer for subnetworks grouped at the levels of **(A)** residue, **(B)** CATH class, **(C)** CATH architecture, **(D)** CATH topology, **(E)** CATH homologous superfamily, **(F)** random sequence suppression, and **(G)** random residue suppression.
(TIFF)

**S6 Fig Mask interpretation of ProtBERT-UR100.** Mean and standard deviation percent of parameters pruned by layer for subnetworks grouped at the levels of **(A)** residue, **(B)** CATH class, **(C)** CATH architecture, **(D)** CATH topology, **(E)** CATH homologous superfamily, **(F)** random sequence suppression, and **(G)** random residue suppression.
(TIFF)

**S7 Fig Mask interpretation of CARP-640M.** Mean and standard deviation percent of parameters pruned by layer for subnetworks grouped at the levels of **(A)** residue, **(B)** CATH class, **(C)** CATH architecture, **(D)** CATH topology, **(E)** CATH homologous superfamily, **(F)** random sequence suppression, and **(G)** random residue suppression.
(TIFF)

**S8 Fig Mask interpretation of Dayhoff-170M-UR90.** Mean and standard deviation percent of parameters pruned by layer for subnetworks grouped at the levels of **(A)** CATH class, **(B)** CATH architecture, **(C)** CATH topology, **(D)** CATH homologous superfamily, and **(E)** random sequence suppression.
(TIFF)

**S1 Table Training and hyperparamter configurations and masked modules for subnetwork learning across four pretrained protein language models.** Each PLM subnetwork differs in which modules are masked, according to the model architecture, following evidence that knowledge is localized in representational modules [22]. Masking is thus applied to self-attention or convolutional projections while leaving embeddings, bias, and normalization layers intact. Subnetworks trained within a single PLM reuse the same hyperparameter configuration shown below. ESM-2 650M, ProtBERT-UR100, and CARP-640M are masked-language-model PLMs trained with the same MLM objective, whereas Dayhoff-170M-UR90 is an autoregressive model trained with a next-token prediction objective.
(PDF)

**S2 Table Mean and standard deviation of ESM-2 CATH Class-level subnetwork masked language modeling performance across multiple seeds.** Three subnetworks were independently trained for each CATH Class suppression target (Mainly Alpha, Mainly Beta, Alpha-Beta) to assess the reproducibility of mask learning given random initialization of mask scores. Reported are the mean ± standard deviation of masked language modeling perplexity for subnetworks and baseline ESM-2 perplexity stratified by categories of inputs.
(PDF)

**S3 Table Overview of subnetwork definitions.** Each subnetwork is trained to selectively suppress either a specific type of residue belonging to a secondary structure or a set of sequences that belong to the same CATH category

classification. In our study, we consider only the top 10 most frequent labels in each CATH category, and report the number of sequences and predominant type of secondary structure content in each category.
(PDF)

**S4 Table ESM-2 650M subnetwork and baseline language modeling performance.** Below we report the mean and standard deviation of the subnetwork and ESM-2 baseline perplexities (illustrated in Fig 2C) stratified by categories of inputs. The *t*-test is performed on paired inputs of the subnetwork performance and ESM-2 baseline performance and *p*-values are reported below. The random residue control subnetwork is trained to suppress random residues, but we independently evaluate and report the MLM performance of this subnetwork on alpha helices and beta sheets. All per-sequence metrics are available in CSVs in the code repository.
(PDF)

**S5 Table ProteinMPNN perplexities on CATH PDBs (Mainly Alpha, Mainly Beta, and Alpha-Beta classes).** Reported values correspond to mean, standard deviation, minimum, and maximum perplexities across all CATH domains within each structural CATH Class.
(PDF)

**S6 Table ESM-2 650M subnetwork and baseline structure prediction performance using the ESMFold (650M) folding trunk on the validation datasets**. For TM-score, RMSD, and pLDDT, we report the mean $\pm$ standard deviation of the subnetwork performance across all sequences within each category of suppression and maintenance inputs. ESM-2 (650M) performance on the same categories is reported as the PLM baseline. To quantify the differences in subnetwork and PLM performance, we perform a paired t-test on all (i) suppression inputs and (ii) maintenance inputs, computing the difference in $\Delta\text{metric} = \text{metric}_{\text{Subnet.}} - \text{metric}_{\text{ESM-2.}}$. We then perform a Kolmogorov–Smirnov (KS) test on these differences to assess whether the distribution of $|\Delta_{\text{metric,supp}}|$ is significantly greater than that of $|\Delta_{\text{metric,maint}}|$. Our evaluation scheme is illustrated in Fig 3A. We report *p*-values for both statistical tests for each subnetwork; significant *p*-values are in bold. For residue-level suppression, we evaluate structure prediction on CATH class categories of mainly alpha and mainly beta sequences as a proxy for evaluating alpha helix and beta strand performance. We report performance on the only residue-specific metric of pLDDT in Fig 3F and 3G. The random residue suppression subnetwork, i.e. residue-control, is one subnetwork but we evaluated it separately on alpha helices and beta sheets. All per-sequence structure prediction metrics are provided via CSVs in our code repository.
(PDF)

## Author contributions

**Conceptualization:** Ria Vinod, Lorin Crawford, Kevin K. Yang.

**Data curation:** Ria Vinod.

**Formal analysis:** Ria Vinod.

**Investigation:** Ria Vinod.

**Methodology:** Ria Vinod, Ava Amini, Kevin K. Yang.

**Project administration:** Ria Vinod.

**Resources:** Lorin Crawford.

**Software:** Ria Vinod.

**Supervision:** Ava Amini, Lorin Crawford, Kevin K. Yang.

**Validation:** Ria Vinod.

**Visualization:** Ria Vinod, Ava Amini.

**Writing – original draft:** Ria Vinod, Lorin Crawford, Kevin K. Yang.

**Writing – review & editing:** Ria Vinod, Ava Amini, Lorin Crawford, Kevin K. Yang.

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
