## [Decision Letter · Decision Letter 0]

21 Sep 2025

PCOMPBIOL-D-25-01366

Trainable subnetworks reveal insights into structure knowledge organization in protein language models

PLOS Computational Biology

Dear Dr. Yang,

Thank you for submitting your manuscript to PLOS Computational Biology. After careful consideration, we feel that it has merit but does not fully meet PLOS Computational Biology's publication criteria as it currently stands. Therefore, we invite you to submit a revised version of the manuscript that addresses the points raised during the review process.

Please submit your revised manuscript within 60 days Nov 21 2025 11:59PM. If you will need more time than this to complete your revisions, please reply to this message or contact the journal office at ploscompbiol@plos.org. Please include the following items when submitting your revised manuscript:

We look forward to receiving your revised manuscript.

Kind regards,

Rachel Kolodny

Academic Editor

PLOS Computational Biology

Nir Ben-Tal

Section Editor

PLOS Computational Biology

**Additional Editor Comments:**

Reviewer #1:

Reviewer #2:

**Journal Requirements:**

At this stage, the following Authors/Authors require contributions: Kevin K Yang. Please ensure that the full contributions of each author are acknowledged in the "Add/Edit/Remove Authors" section of our submission form.

5) We have noticed that you have uploaded Supporting Information files, but you have not included a list of legends. Please add a full list of legends for your Supporting Information files after the references list.

**Reviewers' comments:**

Reviewer's Responses to Questions

**Comments to the Authors:**

Reviewer #1: Vinod et al. present a method demonstrating how structural information is factorized within the large, sequence-based protein language model ESM2 using trainable subnetworks. By comparing both perplexity changes within ESM2 and alterations in structure prediction performance with ESMFold, the authors provide clear evidence that ESM2 encodes structural elements into distinct subnetworks. Masking these subnetworks significantly perturbs protein structure predictions, underscoring the validating the approach.

The study provides valuable a framework to interpret pLMs to gain insights into the internal workings and limitations. Overall, the training procedures, datasets, and evaluation methodologies are clearly described and executed. However, the manuscript would benefit from additional clarity and development of certain key points (see below). The authors have commendably shared both their code and the training dataset.

Major:

- It is unclear why applying the unmodified ESMFold trunk to altered embedding spaces of the subnetwork remains a valid evaluation approach. Although including subnetworks trained to suppress random inputs partially addresses this concern, a discussion or additional analysis (such as an ablation experiment) would help to justify the choice.

- Surprisingly, suppressing random positions had a substantial negative impact on performance, I would have expected that this control would overall maintain performance. A clear discussion of this unexpected outcome and its implications is needed.

- The subnetworks were trained specifically on single-domain proteins annotated with CATH annotations. It would be important to show how subnetworks generalize to multi-domain proteins that combine suppressed and maintained domains.

- Currently it is not possible to install the code. While installing dependencies according to the GitHub instructions, dependency conflicts occurred with Python 3.10 and 3.11 at the step:

error message with python3.10: ERROR: Ignored the following versions that require a different python version: 1.3.0 Requires-Python >=3.11; 1.3.3 Requires-Python >=3.11; 1.4.0 Requires-Python >=3.11; 2.3.0 Requires-Python >=3.11; 2.3.1 Requires-Python >=3.11; 2.3.2 Requires-Python >=3.11; 3.5 Requires-Python >=3.11; 3.5rc0 Requires-Python >=3.11

ERROR: Could not find a version that satisfies the requirement plmprobe==0.1 (from versions: none)

ERROR: No matching distribution found for plmprobe==0.1

error message with python 3.11: ERROR: Ignored the following versions that require a different python version: 1.21.2 Requires-Python >=3.7,<3.11; 1.21.3 Requires-Python >=3.7,<3.11; 1.21.4 Requires-Python >=3.7,<3.11; 1.21.5 Requires-Python >=3.7,<3.11; 1.21.6 Requires-Python >=3.7,<3.11

ERROR: Could not find a version that satisfies the requirement plmprobe==0.1 (from versions: none)

ERROR: No matching distribution found for plmprobe==0.1

All was tested in a conda environment with Python 3.10 as well as 3.11. See the installation instructions:

pip install --extra-index-url https://download.pytorch.org/whl/cu121-renvironments/h100env_requirements.txt

Minor:

- Please briefly explain why only ESM-2 650M was chosen. Showing whether the observed performance degradation generalizes to other pPLMs would strengthen the analysis.

- Currently it is not clear why the MLM maintenance training objective is necessary in addition to the “maintenance goal”. As far as I understand, the maintenance goal has the purpose of keeping performance intact on a set of inputs to be compared with those suppressed in the suppression objective, but naming both objectives “maintenance” makes the latter seem redundant. This needs clarification.

- Figure 2A under the “Perplexity” subtitle there is a tiny red cross inside a square, seems left there by mistake.

- Even though the dataset is not that large, distributing sequences in .fasta format and structures in .pdb format seems a bit inefficient. Please consider providing a compressed version.

Reviewer #2: This study investigates how protein language models (PLMs) internally organise and “factorise” knowledge of protein structural features. It focuses on ESM-2 and introduces trainable subnetworks as a method to probe the model’s learned parameters for structure-specific information. In practice, the authors define suppression sets of sequences belonging to a specific structural class using CATH hierarchical categories and secondary structure labels and maintenance sets for all other sequences. By applying a multi-objective training, they obtain sparse subnetworks of the original model specialised to each structural concept. A total of 36 such subnetworks were trained, covering multiple levels of structure granularity. The authors then assess how removing or isolating these structural knowledge subnetworks affects downstream 3D structure prediction using ESMFold.

The study finds that certain structural features in the PLM are indeed encoded in a factorised manner. The model can predominantly disentangle extremely coarse-grained structural or very fine-grained ones, whereas intermediate-level categories are not cleanly separable. The authors present this subnetwork approach as a new framework to study feature entanglement in biological language models and suggest it could be used to guide models toward more interpretable representations. Overall, this is an original study that provides valuable insights into the modularity of learned representations in pLMs. The paper is a valuable contribution to both the machine learning and protein science communities. However, there are also important limitations and open questions, which should be addressed.

Major Issues

1. A key limitation is that the analysis is performed on only one PLM (ESM-2). While ESM-2 is a high-quality model, the conclusions drawn – that structural knowledge is factorised in certain ways – may not universally hold for other architectures or training methods. For example, a different protein language model (such as ProtT5) might organise information differently. The study would be stronger if the authors at least discussed this limitation thoroughly or, ideally, provided some evidence on an additional model to show whether the phenomenon is consistent. As it stands, readers would automatically assume that the findings reflect properties of all large PLMs, which is an extrapolation not directly tested. This is a significant issue because the claim about how PLMs factorize structural categories could be model-specific.

2. The finding that intermediate structural categories cannot be cleanly disentangled by any subnetwork is intriguing, but the paper does not thoroughly probe why this is the case. This is a major interpretative gap. The authors report the outcome that subnetworks for medium-grained classes don’t achieve the same selective suppression, but they stop short of analysing the causes or implications. Is it because the model inherently entangles those features (perhaps due to overlapping sequence patterns or evolutionary signals)? Or could it be due to the definition of those categories being broad or heterogeneous? Currently, one might feel the authors identified a limitation (PLM knowledge not factorised at mid-level) but did not leverage it to provide deeper insight. Without such exploration, the treatment of this result is a bit superficial. This is an important issue because it touches on the limits of the model, acknowledging and examining it in detail would enhance the scientific soundness of the work.

3. Although the training objective explicitly tries to preserve performance on maintenance (inputs, the results show that even those inputs’ structure predictions were statistically significantly perturbed by the subnetwork masking. This implies that the subnetworks, while largely maintaining perplexity on non-suppressed sequences, still altered the representations in a way that affects downstream tasks broadly. From a methodological standpoint, this is a concern: the intention was to localise and remove only the knowledge relevant to one class, yet it proved impossible to do so without collateral damage to other classes’ predictions. In other words, the PLM’s knowledge of one structural category is not perfectly modular – some of it is shared or intertwined with other categories. The authors do note this outcome, but a criticism is that the paper doesn’t propose or investigate solutions or deeper implications of this entanglement. For example, could adjusting the loss weighting (making the maintenance goal stronger) produce a “cleaner” separation at the expense of less suppression? This issue is significant because it shows a limitation in the subnetwork approach’s efficacy: the subnetworks cannot isolate one concept without affecting others, which slightly undermines the claim that structural categories were found to be factorised. It would be valuable for the authors to discuss this point in depth.

4. The approach used (differentiable masking with a multi-term loss) is technically complex, and the paper provides many details (some delegated to appendices) about hyperparameter choices (mask initialisation, mask update schedule, loss weight λ coefficients, etc.). A potential issue here is the sensitivity and stability of this training procedure. The authors do not thoroughly report on how robust the mask learning is – for instance, if one trains the same subnetwork multiple times from different random initial mask values, do we obtain the same set of important weights, or does it vary? There is a concern that the optimisation problem for finding subnetworks is non-convex and could have multiple solutions (or degenerate solutions where the mask fails to converge to a clear separation). If the success of isolating a given structural class depends heavily on hyperparameter tuning or luck in initialisation, that would weaken the reliability of the conclusions. This is a major technical point because it speaks to the rigour of the method: without demonstrating stability, it’s hard to know how definitive the identified knowledge-critical weights are.

5. The authors choose to apply the trainable mask only to the later layers of the transformer (layers 6–33) based on a prior finding that most structural signals reside in higher layers. While this is a reasonable heuristic to improve training stability, it may introduce a subtle bias. By not allowing any changes to the first 5 layers, the method presupposes that no crucial structural information is encoded there. If that assumption were false, the subnetwork might fail to remove certain knowledge simply because it wasn’t permitted to. Early transformer layers typically capture generic features, but the authors should clarify this choice. Ideally, one could test if allowing the mask over all layers changes the results. In other words, the paper aims to find all weights responsible for a given structural category, yet in practice, it only searches in a subset of the network. If important features were in layers 1–5, they would be missed.

6. The study successfully identifies masks that isolate structural categories, but it provides limited interpretation of which parts of the model those masks correspond to. For instance, readers might be curious if certain transformer layers or attention heads are consistently pruned for particular structural classes. As far as I understand, the article does not report any analysis of the mask patterns. Including even a qualitative summary of mask characteristics would strengthen the connection between the subnetwork and the notion of knowledge organisation. Without it, the subnetworks are abstract. We know they exist and affect performance, but not how or where in the model the structural knowledge was stored. As a suggestion, the authors could visualise or summarise the fraction of weights pruned per layer. Currently, the paper’s focus is on outcomes of mask application; adding interpretability of the mask itself would enrich the contribution.

7. The manuscript cites many relevant studies in the introduction, which is commendable, but in the discussion of results, it could do more to relate its findings back to those studies. It doesn’t affect the validity of the work, but a richer comparison with literature would highlight the study’s novelty and also reassure readers that important previous findings have been considered in interpreting the results.

**Have the authors made all data and (if applicable) computational code underlying the findings in their manuscript fully available?**

Reviewer #1: Yes

Reviewer #2: Yes

PLOS authors have the option to publish the peer review history of their article (what does this mean?). If published, this will include your full peer review and any attached files.

Reviewer #1: No

Reviewer #2: No

**Figure resubmission:**
---

## [Decision Letter · Decision Letter 1]

19 Jan 2026

Dear Yang,

We are pleased to inform you that your manuscript 'Trainable subnetworks reveal insights into structure knowledge organization in protein language models' has been provisionally accepted for publication in PLOS Computational Biology.

Best regards,

Rachel Kolodny

Academic Editor

PLOS Computational Biology

Nir Ben-Tal

Section Editor

PLOS Computational Biology

Reviewer's Responses to Questions

**Comments to the Authors:**

Reviewer #2: I thank the authors for carefully addressing all of my concerns.

**Have the authors made all data and (if applicable) computational code underlying the findings in their manuscript fully available?**

Reviewer #2: Yes

PLOS authors have the option to publish the peer review history of their article (what does this mean?). If published, this will include your full peer review and any attached files.

Reviewer #2: **Yes:** Tunca Doğan, PhD

---

## [Editor Report · Acceptance letter]

PCOMPBIOL-D-25-01366R1

Trainable subnetworks reveal insights into structure knowledge organization in protein language models

Dear Dr Yang,

I am pleased to inform you that your manuscript has been formally accepted for publication in PLOS Computational Biology. Your manuscript is now with our production department and you will be notified of the publication date in due course.

With kind regards,

Anita Estes
